# Association between the Duration of the Active Commuting to and from School, and Cognitive Performance in Urban Portuguese Adolescents

**DOI:** 10.3390/ijerph192315692

**Published:** 2022-11-25

**Authors:** Ana Rodrigues, Hélio Antunes, Ricardo Alves, Ana Luísa Correia, Helder Lopes, Bebiana Sabino, Adilson Marques, Andreas Ihle, Élvio Rúbio Gouveia

**Affiliations:** 1Department of Physical Education and Sport, Faculty of Social Sciences, University of Madeira, 9000-082 Funchal, Portugal; 2Research Center in Sports Sciences, Health and Human Development (CIDESD), 5001-801 Vila Real, Portugal; 3Higher School of Education, Polytechnic Institute of Beja, 7800-295 Beja, Portugal; 4Interdisciplinary Centre for the Study of Human Performance (CIPER), Faculty of Human Kinetics, University of Lisbon, 1499-002 Cruz Quebrada, Portugal; 5Environmental Health Institute (ISAMB), Faculty of Medicine, University of Lisbon, 1649-020 Lisbon, Portugal; 6Department of Psychology, University of Geneva, 1205 Geneva, Switzerland; 7Center for the Interdisciplinary Study of Gerontology and Vulnerability, University of Geneva, 1205 Geneva, Switzerland; 8Swiss National Centre of Competence in Research LIVES—Overcoming Vulnerability: Life Course Perspectives, 1015 Lausanne, Switzerland; 9Laboratory of Robotics and Engineering Systems (LARSYS), Interactive Technologies Institute, 9020-105 Funchal, Portugal

**Keywords:** active commuting to school, cognitive performance, adolescents

## Abstract

This study aimed to analyze the differences between active commuting to school (ACS) and non-ACS in cognitive performance (CP), and the association of ACS duration with CP. This cross-sectional study included 370 adolescents (males *n* = 170), with a mean age of 15.28 ± 2.25 years. CP was assessed through an interview, and ACS, extracurricular physical activity, and socioeconomic status was assessed by self-report. Body composition was assessed using the FitnessGram test battery. One in two adolescents did ACS (51.6%). ACS was associated with boys (53.9%), younger adolescents (14.91 ± 2.15 vs. 15.69 ± 2.29), those having school social support (55.0%), and those doing one or more extracurricular physical activities (53.6%), compared to non-ACS participants (*p* < 0.05). The analysis of covariance, after controlling for age, sex, school social support, and participation in extracurricular physical activity, showed an effect of ACS on the total cognitive score (F_(2,362)_ = 3.304, *p* < 0.05). The CP was higher in adolescents with more than 30 min of ACS than non-ACS (*p* < 0.05). The influence of ACS duration can be seen in the dimensions of inductive reasoning (ß = 0.134, t = 2.587, *p* < 0.05) and working memory (ß = 0.130, t = 2.525, *p* < 0.05). The role of ACS for CP, as well as guidelines for future research, are discussed.

## 1. Introduction

There is consensus among the scientific community on the benefits of physical activity (PA) in health indicators, such as cardiorespiratory and muscular fitness, bone and cardiometabolic health [1], mental health [2], and cognitive function [3]. However, despite the numerous benefits associated with physical activity in pediatric populations, we have recorded a decrease in physical activity levels in childhood and adolescence with a high number of children and young people who do not meet the daily recommendations of physical activity [4].

The development of studies on the influence of PA on mortality and morbidity factors in the medium and long term has been one of the scientific community’s concerns in outlining intervention strategies. One rapidly growing health problem is dementia, affecting approximately 50 million people worldwide [5]. The World Health Organization predicts that, by 2050, the incidence of new cases each year will be 30 million, roughly three times the current rate [5]. In this context, the development of studies on cognitive functioning and its relationship with factors such as PA in the elderly [6] and adults, but also in children [7], has aroused interest among the scientific community. Scientific evidence suggests that PA improves academic outcomes and cognitive performance in children and adolescents [8,9,10]. This relationship is supported by the fact that PA is associated with a set of structural and functional changes in the brain, which potentiate neuroplasticity [11]. PA characteristics such as duration, intensity, and frequency should be considered in the effects of PA on cognitive functioning [11].

Duration of PA is one of the characteristics most correlated with cognitive function in the literature, and higher levels of PA seem to be associated with gains in cognitive function [3]. Daily PA levels result from several physical activities developed during the day, whether organized (such as physical education or school sports classes) or informal (recreational activities).

Active transport to and from school has proved to be a strategy to enhance daily PA levels among children and young people [10,12,13,14]. However, there is some evidence that this type of activity has declined in recent decades [15,16]. At the same time, there is a growing interest in research related to this topic, with benefits being recognized for public health and reducing pollution caused by motorized transport [17].

Active commuting to school for adolescents and children represents a complex behavior in which multiple factors influence: (i) personal (socio-demographic variables, behavioral patterns; motivations) [18]; (ii) social (company of peers, family resources, school context) [17,18,19,20]; (iii) environmental (area of residence, distance from school and perception of security) [19,20,21,22,23]. Young people who actively commute to school are less likely to have health complaints, especially psychological symptoms [24], and have lower levels of obesity [25,26].

However, despite recognizing the role of PA in the cognitive performance of active transport in some health indicators, few studies have explored the association of active transport and its characteristics in cognitive performance [10]. Aspects inherent to the characteristics of active transport, such as duration, intensity, and interaction with involvement and its relationship with cognitive performance, also seem to be aspects that need investigation. Martinez-Gomez et al. [27] points out that in adolescents urban girls, active commuting to and from school (ACS) lasting more than 15 min per day is positively associated with cognitive performance. Other studies indicate that a duration between 30 to 60 min of ACS is related to high academic performance [28]. Nonetheless, other studies evaluating this relationship and ACS duration do not present consensual results, with the benefits of ACS being reported on cognitive performance [12,28,29], As well as the lack of association between ACS and cognition and academic results [10,30].

Therefore, to address these major gaps, the purpose of the study was: (i) to study the differences between young people who carry out active and passive transport to and from school in cognitive performance, and (ii) to analyze the association of the duration of active transport with different components of cognitive performance (prospective memory, short-term memory, working memory, verbal fluency, inductive reasoning, and long-term verbal memory).

## 2. Materials and Methods

### 2.1. The Sample

The study included 370 subjects of both sexes (male = 170) aged between 11 and 20 years old (15.28 ± 2.25), from middle school (*n* = 145) and high school students (*n* = 225) who were residents in urban areas. This study includes all participants of the research project “Physical Education in Schools from the Autonomous Region of Madeira” (EFERAM-CIT; https://eferamcit.wixsite.com/eferamcit (accessed on 1 October 2022). The participants were from 5 schools in Funchal (capital of the island of Madeira).

Participants were informed about the objectives of the study, and written informed consent was obtained from their legal guardians. The study received ethical approval from the Scientific Committee of the Faculty of Physical Education and Sports at the University of Madeira (Reference: ACTA N.77–12.04.2016). This study was also approved by the Regional Secretary of Education and the school’s headmaster.

The evaluations were conducted by graduates in Physical Education and Sport. The evaluators received education and training, with: (i) instructions and demonstrations for the CP, questionnaires, and body composition; (ii) the field team members practiced on each other; and (iii) the team participated in a pilot study, for which all variables were assessed in 8 boys and 7 girls aged 16–18 years. The adolescents were evaluated twice at an interval of 1 week. This pilot study indicated good to acceptable test–retest reliabilities for all assessments (interclass correlation coefficient between 0.797 and 0.999).

The assessments lasted 45 min (one session), including an interview to assess cognitive function, body composition, and questionnaire (demographic, socioeconomic, and physical activity profile).

### 2.2. Instruments

#### 2.2.1. Commuting to School

The self-reported questionnaire measured patterns of ACS. Two questions were asked about the mode and duration of commuting to school: (1) “How do you usually travel from home to school and from school to home?”—response options were walking, biking, bus, car, and motorcycle; and (2) “How long does it usually take you to travel from home to school or from school to home?”—response in minutes. If the adolescents reported at least one of the trips as active (home to school or school to home), they were included in the active commuting group (ACS group). The subjects were included in the non-active commuting group (non-ACS group) if they did not report at least one of the trips as active. Additionally, participants were categorized in 3 categories for the duration of ACS: (i) Non-ACS (non-active commuting); (ii) ACS ≤ 30 min (active commuting with a maximum duration of 30 min); and (iii) ACS > 30 min (active commuting for more 30 min).

#### 2.2.2. Cognitive Performance (CP)

Cognitive function was assessed through an interview using the Cognitive Telephone Screening Instrument. The Cognitive Telephone Screening Instrument (COGTEL) [31,32,33] included six cognitive tasks: (1) Prospective memory—assessed using an event-based task (i.e., a task in which the execution of the intended action is triggered by the presentation of a specific external cue). The formation of the intention occurred at the beginning of the test interview (0 or 1 point). (2) Verbal short memory—with a verbal paired-associate memory test (immediate recall; 0–8 points). (3) Verbal long-term memory—using the same word pairs as in the verbal short-term memory test in the delayed-retrieval test at the end of the test interview (0–8 points). (4) Working memory—using the backward digit-span (0–12 points). (5) Verbal fluency (i.e., executive functioning)—participants produced as many words as possible that begin with a given letter (0 to unlimited; as many words as the participant can name within 60 s). (6) Inductive reasoning—participants were provided with sequences composed of five numbers that were constructed following a specific mathematical rule (0–8 points). The scores of the six cognitive tasks can be combined into a weighted total score (7.2 * prospective memory + 1.0 * verbal short-term memory + 0.9 * verbal long-term memory + 0.8 * working memory + 0.2 * verbal fluency + 1.7 * inductive reasoning score). Members of the field team administered the questionnaire during face-to-face interviews. A detailed description of the evaluation procedures, namely procedures and scoring, can be found in Ihle et al. [31], Gouveia et al. [32] and Kliegel et al. [33]. Memory is used to describe the cognitive mechanism by which information is acquired, retained, and recalled.

#### 2.2.3. Extracurricular Physical Activity (EPA)

Participation in extracurricular physical activity was self-reported and determined by the following questions: (i) “Do you practice in any type of physical sports activity beyond physical education?” (Yes/No); (ii) “If yes, which sport do you practice?”; and (iii) “If yes, in what context?” (as a school sports athlete/as a federated athlete in an association). Subsequently, the participants were categorized into 3 groups: (1) OPEd—students who have physical education classes as their only organized physical activity; (2) SS—participants who, in addition to physical education, participate in school sports; (3) SC—adolescents registered as athletes in federations.

#### 2.2.4. Socioeconomic Status (SES)

School social support (SSS) is a set of economic support for students according to the household’s annual income and the number of children. Thus, it is a good indicator of the participants’ socioeconomic status. Adolescents were categorized into 4 categories according to academic household income (Table 1). Lower household income corresponds to the lowest social support school bracket.

#### 2.2.5. Body Composition

Weight, height, and adiposity folds (triceps and calf) were evaluated according to the procedures described in the FitnessGram test battery [34].

Body height and weight were measured with the adolescents having bare feet and wearing light underclothes. Height was measured to the nearest 1 mm and body weight to the nearest 0.5 kg using a balance (SECA 761, Hamburg, Germany) and stadiometer (SECA 213, Hamburg, Germany). Body mass index was calculated as body weight in kilograms divided by height in meters squared. Overweight and obesity adolescents were classified according to age- and sex-specific body mass index cutoffs proposed by the FitnessGram test battery [34]. Subcutaneous adiposity folds were evaluated using the Skinfold Caliper (Harpenden, England).

The %body fat was determined based on triceps and calf skinfold [35].

### 2.3. Data Analysis

Descriptive statistics were made to characterize the sample. The continuous variables were expressed as the mean and standard deviation and frequency distribution for categorical data. Statistical normality was tested using the Kolmogorov–Smirnov test.

To study the differences between non-ACS vs. ACS, we used Pearson χ^2^ and the Student *t* test in nominal (e.g., sex) and qualitative variables with a normal distribution (e.g., height, weight), respectively.

To examine the difference between commuting to school (non-ACS, ACS < 30 min, and ACS > 30 min) in cognitive performance levels, we conducted an analysis of covariance after controlling for age, sex, school social support, and participation in extracurricular physical activity. The post hoc Bonferroni test was used to determine in which of the 3 groups there were differences.

Multivariate analysis of variance with the post hoc by the Bonferroni test was used to determine the differences between the group of ACS in dimensions of cognitive performance after controlling for age, sex, school social support, and participation in extracurricular physical activity.

Multiple linear regression was used to determine the influence of active transport duration on cognitive performance dimensions.

Statistical analyses were performed using SPSS version 27.0 statistical software for Windows (SPSS Inc., Chicago, IL, USA). The significance level adopted was 5%.

## 3. Results

Table 2 shows the adolescents’ characteristics. Approximately half of the sample (51.6%) reported performing ACS at least one way to and from school. On average, the journey takes 26.47 ± 15.40 min per day. Approximately half of the participants do not have school social support, which indicates a medium to high SES. One in five adolescents is overweight, and 8.4% are obese. One in two participants have physical education classes as their only organized PA; 12.4% participate in school sports, and 38.9% practice a sport in a club.

Participants with ACS compared to non-ACS are mostly boys (53.9%), younger (14.91 ± 2.15 vs. 15.69 ± 2.29), have school social support (55.0%), and with one or more extracurricular organized physical activities (53.6%) (*p* < 0.05). There are no differences in nutritional status (*p* > 0.05).

There is an effect of ACS on the total cognitive score (F_(2,362)_ = 3.304, *p* = 0.038) after controlling for the impact of sex, age, school social support, and EPA. Cognitive performance was higher in adolescents with more than 30 min of ACS than in non-commuters (*p* < 0.05) (Figure 1).

The total cognitive function, composed of six dimensions combined into a weighted total score, was explored for the differences between the three groups of ACS. In all six dimensions, participants with ACS > 30 min presented higher values than the non-ACS and ACS < 30 groups. However, they only have statistical significance in the working memory and inductive reasoning dimensions after controlling for sex, age, school social support, and EPA. After adjustment by the Bonferroni test, significant differences were found only between non-ACS and ACS > 30 min (Table 3).

Multiple linear regression was used to determine the influence of ACS duration, EPA, age, sex, and school social support on the working memory and inductive reasoning dimensions. The independent variables explain 4.3% of the variability of intuitive reasoning (F_(3,366)_ = 5.494, *p* = 0.001, R^2^ = 4.3%), the ACS explains 1.2% (ß = 0.134, t = 2.587, *p* < 0.05), age 1.5% (ß = 0.161, t = 2.984, *p* < 0.05), and EPA 1.6% (ß = 0.131, t = 2.461, *p* < 0.05). ACS duration is the only independent variable that was included in the regression model (F_(1,368)_ = 6.375, *p* = 0.012, R^2^ = 1.7%), explaining 1.7% of working memory variability (ß = 0.130, t = 2.525, *p* < 0.05).

## 4. Discussion

Regularly, PA has been associated with physical and social health in children and adolescents. Active transport to and from school is one of the daily opportunities to increase PA levels in these age groups [36]. Our study shows a higher percentage of ACS compared to other studies carried out in Portugal [37]. However, our participants reside in an island region, which may explain this percentage difference [38]. The characteristics of environment such as proximity between home and school, security [22], population density [39], and built [40] and natural environment [41] are also factors to be considered; however, this association needs further investigation.

The relationship between ACS and CP is reported in the pediatric population in other studies [10,12,27,28,42]. One of the main results of this study is that participants who perform active transportation have better CP, independent of potential confounding factors such as gender, age, school social support, and EPA. The influence of factors such as gender, age, SES, obesity, physical fitness and PAE, on the ACS, on the CP, and on the relationship between both is reported in the literature [10]. It was found that participants reporting ACS are mostly male, on average younger, with SSS, and with one or more PAE, compared to non-ACS participants. This study was controlled for several covariates: age, gender, SSS and EPA. This is supported by the fact that CP is influenced by these variables and there are differences between ACS and non-ACS.

The duration of the ACS seems to be of particular importance. The scientific community has difficulties in defining an ACS duration that causes beneficial effects on cognitive functioning. In our study, the benefits are particularly significant among participants who perform more than 30 min of ACS daily. However, beneficial effects are also associated with shorter duration ACS (15 min), particularly in girls [27]. García-Hermoso et al. [28], by evaluating four groups (Non-ACS, ACS < 30 min, ACS 30–60 min and ACS > 60 min), found that academic performance was higher on average among participants reporting ACS between 30–60 min. The other three groups had identical results (*p* > 0.05) [28]. This result highlights the need to analyze more characteristics of the ACS beyond the simple duration indicator. Course characteristics and intensity of ACS should be aspects to be considered in the relationship between ACS and CP. However, the contribution of ACS to daily PA levels is undeniable [12]. Several studies demonstrate an association between PA and cognitive performance [7,8] supported by the structural and functional changes that occur with physical activity in the brain [11]. Among some of the benefits of physical activity at the cognitive level is the increased grey matter volume in frontal and hippocampal regions [7], the increased level of neurotrophic factors, blood flow, stimulation of neurogenesis, and increased neural transmission in the hippocampus [11]. The brain changes appear to be associated with a greater ability to concentrate, memory, and, consequently, better academic performance, particularly in children.

However, the benefits of physical activity on cognitive performance seem to be associated with vigorous intensity [7,8], which is not the case with ACS (which is often associated with moderate physical activity). In the study, ACS assessment was used through self-report, essential for more studies about active transport, through objective instruments, which allow exploration of duration and intensity. This aspect is significant, considering the physical context in which the participants developed the ACS, with large gaps accumulated over short distances.

In the dimensions of CP, working memory and intuitive reasoning were associated with the duration of ACS, controlling for sex, age, socioeconomic support, and EPA. The benefits of PA on cognitive ability is reflected in improvements in learning and memory tasks in active children compared to sedentary children [16]. Active transport influences working memory, being the dimension of cognitive function where this effect is most evident.

The influence of environmental factors on the structure and physiological functions of the central nervous system is reported by Mandolesi et al. (2017) [11], suggesting that exposure to stimuli and experiencing a situation can increase neuroplasticity, improve cerebrovascular health, and determines the benefits of glucose and lipid metabolism that transport energy resources to the brain.

The influence of the ACS in the working memory dimensions and in inductive reasoning could be the result of the sensory stimulations and of the cognitive demands associated with the interaction with the stimuli during the course [27]. Given the benefits associated with ACS, the high number of non-ACS adolescents is worrying. The values of this study are in line with other studies [17], the ACS seems to be influenced by factors such as sex, age, socioeconomic status, characteristics of involvement, family characteristics, and dynamics [20]. In this context, it is imperative to develop intervention programs to promote ACS, especially considering that most adolescents do not reach the recommended values of physical activity [4].

The study has several limitations. This study is a cross-sectional design, which limits the ability to draw conclusions about the causal direction of the associations. ACS was assessed by self-report; future research should use instruments that quantify duration more objectively (e.g., Global Positioning System). The intensity of PA (ACS and EPA) was not quantified. This may be relevant, as several studies report the influence of PA intensity on CP. The use of more objective PA assessment instruments is pertinent in the development of future studies. The sample only corresponds to five schools in the Autonomous Region of Madeira, which makes it impossible to generalize the results. Furthermore, the study was carried out in an island region of Portugal. The island of Madeira has unique characteristics of environment, namely natural, with significant differences in level over short distances and educational policies that support travel by bus; aspects that certainly influenced the option or not for active transport. Thus, the characterization of involvement and its relationship with active transport and cognitive performance are aspects to be further investigated in future investigations.

## 5. Conclusions

This study highlights the relationship between the duration of ACS and cognitive function. The effects of the ACS seem to be more evident when the duration of the ACS is equal to or greater than 30 min daily. However, further investigation is needed to determine whether the results are supported by increased levels of daily physical activity or by the interaction with engagement in performing ACS. The working memory and inductive reasoning dimensions are shown to be associated with ACS, regardless of sex, age, SSS, and EPA. These data seem to support the influence of interaction with involvement when partaking in ACS.

With the development of the study, several questions emerged that need to be deepened in future investigations: Is a longer duration of the ACS and, consequently, the daily levels of PA alone, necessary for improving cognitive performance? Is it the subject’s interaction with the involvement and its stimuli that enhances cognitive performance? Or a combination of both? Thereby, the present study may stimulate future research to scrutinize the mechanisms involved in more detail.

## Figures and Tables

**Figure 1 ijerph-19-15692-f001:**
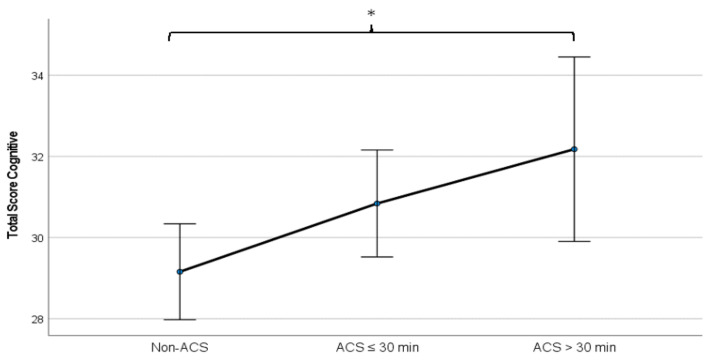
The mean differences in total cognitive score by categories of ACS. Non-ACS—no active communing school; ACS ≤ 30 min—active commuting to school ≤ 30 min per day; ACS > 30 min—active commuting to school > 30 min per day; * *p* < 0.05.

**Table 1 ijerph-19-15692-t001:** Annual income by school social support group.

Group SSS	1 Son	2 Sons	3 Sons	4 Sons	5 Sons
SSS 1	<6204.80 €	<9307.20 €	<12,409.60 €	<15,512.00 €	<18,614.40 €
SSS 2	Between 6204.80 € and 12,409.60 €	Between 930,720 € and 18,614.40 €	Between 2409.60 € and 24,819.20 €	Between 15,512.00 € and 31,024.00 €	Between 18,614.40 € and 37,228.80 €
SSS 3	Between 12,409.60 € and 18,614.40 €	Between 18,614.40 € and 27,921.60 €	Between 24,819.20 € and 37,228.80 €	Between 31,024.00 € and 46,536.00 €	Between 37,228.80 € and 55,843.20 €
SSS 4	≥18,614.40 €	≥27,921.60 €	≥37,228.80 €	≥46,536.00 €	≥55,843.20 €

Source: Regional Directorate Planning Resources Infrastructure (2022).

**Table 2 ijerph-19-15692-t002:** The adolescents’ characteristics (*n* = 370).

		Total
Sex	Men, *n* (%)	170 (45.9%)
	Women, *n* (%)	200 (54.1%)
Age	Age (years)	15.28 ± 2.25
Scholarly	Middle school, *n* (%)	145 (39.2%)
	High school students, *n* (%)	225 (60.8%)
School social support (SSS)	SSS 1, *n* (%)	76 (20.5%)
	SSS 2, *n* (%)	88 (23.8%)
	SSS 3, *n* (%)	28 (7.6%)
	SSS 4, *n* (%)	178 (48.1%)
Body composition	Height (cm)	162.84 ± 8.75
	Weight (kg)	59.06 ± 13.81
	BMI (kg.m^−2^)	22.15 ± 4.25
	Overweight, *n* (%)	74 (20.0%)
	Obesity, *n* (%)	31 (8.4%)
	% Body fat, (%)	26.12 ± 10.40
Physical activity profile	Only physical education, *n* (%)	180 (48.6%)
	Sports school, *n* (%)	46 (12.4%)
	Sports club, *n* (%)	144 (38.9%)
Active commuting to school	ACS, *n* (%)	191 (51.6%)
	No-ACS, *n* (%)	179 (48.4%)
	Duration (min.day)	26.47 ± 15.40
	ACS ≤ 30 min/day, *n* (%)	143 (38.6%)
	ACS > 30 min/day, *n* (%)	48 (13.0%)
Cognitive performance	Prospective memory	0.53 ± 0.50
	Verbal short memory	4.81 ± 1.52
	Working memory	5.86 ± 1.68
	Verbal fluency	23.37 ± 6.75
	Inductive reasoning	4.15 ± 1.99
	Verbal long-term memory	5.76 ± 1.58
	Total score	30.20 ± 8.05

Values are means (standard deviations ± SD) and number and proportions (%) for categorical data; ACS—active commuting to school; No-ACS—no active commuting to school.

**Table 3 ijerph-19-15692-t003:** Difference in dimension of cognitive performance by categories of ACS.

Cognitive Performance	Total	Non-ACS	ACS ≤ 30	ACS > 30	*p*
Prospective memory	0.53 ± 0.5	0.54 ± 0.50	0.50 ± 0.50	0.55 ± 0.5	0.856
Verbal short memory	4.81 ± 1.52	4.61 ± 1.56	4.98 ± 1.41	5.00 ± 1.62	0.151
Working memory	5.86 ± 1.58	5.72 ± 1.73	5.87 ± 1.58	6.40 ± 1.72	0.016
Verbal fluency	23.37 ± 6.75	23.51 ± 7.03	23.08 ± 6.511	23.71 ± 6.53	0.477
Inductive reasoning	4.14 ± 1.99	3.19 ± 2.01	4.34 ± 1.94	4.44 ± 2.05	0.037
Verbal long-term memory	5.76 ± 1.58	5.59 ± 1.65	5.92 ± 1.38	5.94 ± 1.82	0.297

Non—ACS—Non active communing school; ACS ≤ 30 min—active commuting to school ≤ 30 min for day; ACS > 30 min—active commuting to school > 30 min for day.

## Data Availability

The data presented in this study are available on request from the corresponding author.

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
