# Peer review of "Association between the Duration of the Active Commuting to and from School, and Cognitive Performance in Urban Portuguese Adolescents"

_ijerph, 2022, doi:10.3390/ijerph192315692_

Round 1

Reviewer 1 Report

Thank you for inviting me to review this manuscript.This manuscript examined the impact of the duration of the Active Commuting to and from school on cognitive performance, and the research results have certain theoretical and practical significance. While there are several issues to be solved. First, the association between the duration of the Active Commuting to and from school, and Cognitive Performance are not well established in the introduction part. Thus, the discussion part did not explain the results very well. This could be enhanced. Second, there are some There are some small mistakes that the authors need to proofread carefully. I also made a yellow mark in the manuscript. Finally, as for the format of references, some journals use abbreviations and some do not.

Author Response

Comments and Suggestions for Authors:

Reviewer: First, the association between the duration of the Active Commuting to and from school, and Cognitive Performance are not well established in the introduction part. Thus, the discussion part did not explain the results very well. This could be enhanced. Second, there are some There are some small mistakes that the authors need to proofread carefully. I also made a yellow mark in the manuscript. Finally, as for the format of references, some journals use abbreviations, and some do not.

Answer: We are thankful to the reviewer for the positive comments that contributed to improving the quality of the article.

  • Reviewer

First, the association between the duration of the Active Commuting to and from school, and Cognitive Performance are not well established in the introduction part.

Answer: The association between Active committing to and from school and cognitive performance was reinforced in the introduction (lines 61 – 66 and 81-95). Also, the influence of duration on cognitive performance as an important part of PA was included (lines 89-95).

  • Reviewer

Thus, the discussion part did not explain the results very well. This could be enhanced.

Answer: The discussion was deepened, exploring the following topics: (i) the Prevalence of ACS, confrontation with other studies and exploring possible explanations; (ii) factors such as gender, age, socioeconomic status, and extracurricular physical activity and their influence on ACS, physical activity and the relationship between both (ACS and physical activity); (iii) the influence of the duration of the ACS on cognitive performance, and (iv) possible influences of the environment on the ACS and its relationship with cognitive performance. (lines 383 a 400)

  • Reviewer

Second, there are some There are some small mistakes that the authors need to proofread carefully. I also made a yellow mark in the manuscript.

Answer: Throughout the text, minor corrections were made, namely in terms of abbreviations and some presentation of results.

  • Reviewer

I also made a yellow mark in the manuscript. Finally, as for the format of references, some journals use abbreviations, and some do not.

Answer: All references have been revised accordingly.

(line 378-479)

Reviewer 2 Report

Thank you very much for allowing me to read this interesting manuscript. The manuscript is based on a topic of great interest. I appreciate a well-written paper with a strong background. Nevertheless, there are some comments and recommendations that I would like to make:

(Abstract) Line 32. “PC” Is correct?

Sample: Were all participants in the project "Physical Education in the Schools of the Autonomous Community of Maderieda" recruited and was the sample size estimated beforehand or was there a background study?

Who was the professional who collected the information, the researcher or the teacher? How much time was taken to collect the information from each participant?

Was the information collected in one or more than one session, was the participant or his/her legal guardian assessed?

Regarding the characteristics of the sample, did you take into consideration whether it was a rural/urban population, did all participants live in the same place where they went to school?

The authors have subdivided the sample several times, but it is not very clear in some cases why, they have taken into account characteristics such as gender or socioeconomic status, however these data are not part of the discussion, it would be interesting to know if there is any relationship between physical activity, cognition and these aspects.

Paragraph 198-201 is not clear, please rephrase.

Table 3 does not include the total score. Morveover, could be interesting to know the range of scores and the odd ratio

Line 221. Acronyms should first be included in full text to make them understandable.

Please, separed foot notes from the next paragraph.

The discussion should be enriched.

The paragraph between the lines (237-243) is interesting but not of particular relevance to this study, because no neuroimaging screening have been used.

Is the statement in the first paragraph 228-229 really supported by these results? 

The study has no limitations, please include them.

Author Response

Comments and Suggestions for Authors:

We appreciate the generally positive assessment done by the reviewer, and we feel that the comments and suggestions significantly contributed to improving the quality of this paper. We also appreciate the attention and care given to the analysis. With the development of this work, we intend to explore the benefits of active commuting in cognitive performance, a topic that we consider necessary to deepen the investigation.Regarding the comments and suggestions mentioned:

  • Reviewer: (Abstract) Line 32. “PC” Is correct?

Answer: The reviewer is incorrect. Instead of “PC,” there should be “CP.” The correction was made in the text (line 33).

  • Reviewer: Sample: Were all participants in the project "Physical Education in the Schools of the Autonomous Community of Madeira" recruited and was the sample size estimated beforehand or was there a background study?

Answer: This study includes all participants of the research project “Physical Education in Schools from the Autonomous Region of Madeira” (EFERAM-CIT; https://eferamcit.wixsite.com/eferamcit). The sample is the convenience with students from 5 schools in Funchal (the capital of the island of Madeira). (lines 98-101)

  • Reviewer: Who was the professional who collected the information, the researcher, or the teacher? How much time was taken to collect the information from each participant?

Answer: The evaluations were carried out by graduates in Physical Education and Sport. The evaluators received education and training, with: (i) instructions and demonstrations for the CP, questionnaires, and body composition, (ii) the field team members practiced on each other, and (iii) the team participated in a pilot study, for which all variables were assessed in 8 boys and 7 girls aged 16–18 years. The adolescents were evaluated twice with an interval of 1 week. This pilot study indicated good to acceptable test–retest re-liabilities for all assessments (interclass correlation coefficient between .797 and .999). The assessments lasted 45 minutes (one session), including interview to assess cognitive function, body composition and questionnaire (demographic, socioeconomic and physical activity profile). (line 108-118)

  • Reviewer: Was the information collected in one or more than one session, was the participant or his/her legal guardian assessed?

Answer: The evaluations took place in a single session; the legal guardian was not evaluated, only the adolescents. 

  • Reviewer: Regarding the characteristics of the sample, did you take into consideration whether it was a rural/urban population, did all participants live in the same place where they went to school?

Answer: Considering their area of residence, All participants came from urban areas,. (line 99)

  • Reviewer: The authors have subdivided the sample several times, but it is not very clear in some cases why, they have taken into account characteristics such as gender or socioeconomic status, however these data are not part of the discussion, it would be interesting to know if there is any relationship between physical activity, cognition and these aspects.

Answer: The presentation of results, such as sex, school social support, and age, is due to the influence of these factors on cognitive performance and physical activity, as already reported in the literature. However, we agree with your comment. Thus, given the importance of these parameters in the relationship between cognitive performance and physical activity, we reinforce those relationships between variables in the discussion (lines 257 - 267).

  • Reviewer: Paragraph 198-201 is not clear, please rephrase. “Participants with ACS are mostly boys (53.9%), younger (14.91±2.15 vs. 15.69±2.29), have school social support (55.0%), and with one or more extracurricular organized physical activities (53.6%), compared to non-ACS (p<.05).”

Answer: The sentence was rewritten as follows: Participants with ACS compared to non-ACS are mostly boys (53.9%), younger (14.91±2.15 vs. 15.69±2.29), have school social support (55.0%), and with one or more extracurricular organized physical activities (53.6%) (p<.05).” (lines 215-219).

  • Reviewer: Table 3 does not include the total score. Moreover, could be interesting to know the range of scores and the odd ratio

Answer: Multivariate analysis of variance with the post hoc by the Bonferroni test was used to determine the differences between the group of ACS in dimensions of cognitive performance after controlling for age, sex, school social support, and participation in extracurricular physical activity. We tried to present a clear, synthetic table with essential and perceptible information about the differences between the 3 groups in the different dimensions of cognitive performance. Considering that the total score is the product of the remaining dimensions, it was not included, to explore each dimension.

  • Reviewer: Line 221. Acronyms should first be included in full text to make them understandable.

Answer: the correction of acronyms from AFEO to EPA was made. (lines 240 and 242).

  • Reviewer: Please, separate foot notes from the next paragraph.

Answer: the change was done accordingly.

  • Reviewer: The discussion should be enriched.

Answer: We agree with the reviewer. The discussion was deepened, exploring: (i) the Prevalence of ACS, confrontation with other studies and exploring possible explanations; (ii) factors such as gender, age, socioeconomic status, and extracurricular physical activity and their influence on ACS, physical activity, and the relationship between both (ACS and physical activity); (iii) the influence of the duration of the ACS on cognitive performance, and (iv) possible influences of the environment on the ACS and its relationship with cognitive performance. (lines 248-279).

  • Reviewer: The paragraph between the lines (237-243) is interesting but not of particular relevance to this study, because no neuroimaging screening have been used.

Answer: With reference to these studies, our objective is to understand and keep clear how physical activity causes changes in brain functioning and morphology and how it is associated with cognitive performance and its different dimensions.

  • Reviewer: Is the statement in the first paragraph 228-229 really supported by these results? 

Answer: considering the reviewer's comment, we have changed the sentence“One of the main results of this study is that participants who perform active transportation have better cognitive performance, independent of potential confounding factors such as gender, age, school social support, and PAE.” (line 257-259)

  • Reviewer: The study has no limitations, please include them.

Answer: We agree with the comment, the limitations associated with the study were introduced in the paper (line 316-330).

Reviewer 3 Report

It was a pleasure to review this paper which evaluates the association between the duration of the Active Commuting to 2 and from school, and Cognitive Performance in Urban Portu-3 guese Adolescents. The method is clearly explained. The results, although somewhat sparse, are adequately represented. However, I consider that a literature review section is necessary since there are many relevant references on the subject that have not been cited and a more robust and well-founded conclusion is lacking.

Author Response

Comments and Suggestions for Authors:

We appreciate the comments and suggestions, which will greatly contribute to improving the quality of the paper.

Regarding the comments and suggestions mentioned:

Reviewer:

It was a pleasure to review this paper which evaluates the association between the duration of the Active Commuting to 2 and from school, and Cognitive Performance in Urban Portuguese Adolescents. The method is clearly explained. The results, although somewhat sparse, are adequately represented. However, I consider that a literature review section is necessary since there are many relevant references on the subject that have not been cited and a more robust and well-founded conclusion is lacking.

Answer: The discussion was deepened, exploring: (i) the Prevalence of ACS, confrontation with other studies, and exploring possible explanations; (ii) factors such as gender, age, socioeconomic status, and extracurricular physical activity, and their influence on ACS, physical activity and the relationship between both (ACS and physical activity); (iii) the influence of the duration of the ACS on cognitive performance, and (iv) possible influences of the environment on the ACS and its relationship with cognitive performance. (lines 248-279).

The introduction attempted to deepen the relationship between cognitive performance and active commuting (lines 77-88).

Recent references and reference journals were used. Systematic reviews and meta-analyses are also privileged, as they allow a more global view of the subject.

Round 2

Reviewer 1 Report

There is no big problem. Please make careful revisions according to the Journal's requirements.

Author Response

Dear Reviewer,

Thanks for your positive feedback on the paper. As requested, we revised and followed all the Journal's requirements. Many thanks for your availability to participate in this revision process. Your notes during the revision process were very important and helpful for improving the manuscript.

Reviewer 2 Report

The Authors provided satisfactory revisions to the article.

Table 3. I recommend highlighting the statistically significant results in order to improve the understanding.

Author Response

Dear Reviewer,

Thanks for your positive feedback on the paper. As requested, we highlighted the statistically significant results in table 3. It looks much better, and we believe it increases the understanding of table 3. Many thanks for your availability to participate in this revision process. Your notes during the revision process were very important and helpful for improving the manuscript.

Reviewer 3 Report

The author has made the suggested changes

Author Response

Dear Reviewer,

Thanks for your positive feedback on the paper. Especially, many thanks for your availability to participate in this revision process. Your notes during the revision process were very important and helpful for improving the manuscript.